# Effect of Electron Donating Functional Groups on Corrosion Inhibition of J55 Steel in a Sweet Corrosive Environment: Experimental, Density Functional Theory, and Molecular Dynamic Simulation

**DOI:** 10.3390/ma12010017

**Published:** 2018-12-21

**Authors:** Ambrish Singh, Kashif R. Ansari, Mumtaz A. Quraishi, Hassane Lgaz

**Affiliations:** 1School of Materials Science and Engineering, Southwest Petroleum University, Chengdu 610500, Sichuan, China; 2State Key Laboratory of Oil and Gas Reservoir Geology and Exploitation, Southwest Petroleum University, Chengdu 610500, Sichuan, China; 3Centre of Research Excellence in Corrosion, Research Institute, King Fahd University of Petroleum and Minerals, Dhahran 31261, Saudi Arabia; Ka3787@gmail.com (K.R.A.); maquraishi.apc@iitbhu.ac.in (M.A.Q.); 4Department of Applied Bioscience, College of Life & Environment Science, Konkuk University, 120 Neungdong-ro, Gwangjin-gu, Seoul 05029, Korea; lgaz.hassane@gmail.com

**Keywords:** corrosion, XPS, J55 steel, carbon dioxide, molecular dynamic simulation

## Abstract

Benzimidazole derivatives were synthesized, characterized, and tested as a corrosion inhibitor for J55 steel in a 3.5 wt % NaCl solution saturated with carbon dioxide. The experimental results revealed that inhibitors are effective for steel protection, with an inhibition efficiency of 94% in the presence of 400 mg/L of inhibitor. The adsorption of the benzimidazole derivatives on J55 steel was found to obey Langmuir’s adsorption isotherm. The addition of inhibitors decreases the cathodic as well anodic current densities and significantly strengthens impedance parameters. X-ray photoelectron spectroscopy (XPS) was used for steel surface characterization. Density functional theory (DFT) and molecular dynamic simulation (MD) were applied for theoretical studies.

## 1. Introduction

Carbon dioxide corrosion is the most commonly faced problem in the petroleum industry, and thus it has been a hot research topic for many years. Corrosion prevention consists of different methods, but one of these methods, the introduction of organic compounds as corrosion inhibitors, is both effective and cheap [1,2,3].

The inhibition action of organic compounds depends on the nature of the molecular structure, inhibitor planarity, electron donating functional groups, non-bonding electrons on heteroatoms, i.e., oxygen, nitrogen, and sulfur, and presence of π bonds in the aromatic ring [4]. In recent years, corrosion scientists have been interested in finding green and environment-friendly inhibitors [5]. Benzimidazole derivatives that show antitumor and antimicrobial activities have been categorized in the group of green compounds [6,7].

A survey of the literature reveals that in the past decades, various imidazole and benzimidazole derivatives have been used as anti-sweet corrosion inhibitors [8,9,10,11,12,13,14,15]. However, no literature exits about using benzimidazole as a corrosion inhibitor in a brine solution saturated with carbon dioxide. Benzimidazole has planar structure with two nitrogen atoms that provides a closer approach for interaction with the metal surface, aromatic properties, and an option for introducing different substituents.

The main purpose of the present paper was to elucidate the corrosion inhibition effect of the number of electron donating methoxy groups on the phenyl ring of the three synthesized benzimidazole derivatives, namely, 2-(3,4,5-Trimethoxyphenyl)-1*H*-benzo[*d*] imidazole (TMI), 2-(3,4-Dimethoxyphenyl)-1*H*-benzo[*d*] imidazole (DMI), and 2-(4-Methoxyphenyl)-1*H*-benzo[*d*] imidazole (MMI), for J55 steel saturated with CO_2_ in a 3.5% NaCl solution. Corrosion inhibition properties of the benzimidazole derivatives were analyzed using the static weight-loss method and electrochemical methods, i.e., impedance spectroscopy (EIS) and potentiodynamic polarization. Meanwhile, the J55 steel surface was examined by scanning electron microscope (SEM) and X-ray photoelectron spectroscopy (XPS). The potential site for protonation was estimated using density functional theory (DFT). The interaction of the benzimidazole derivatives with the J55 steel surface was studied by molecular dynamic simulation (MD).

## 2. Experiment

### 2.1. J55 Steel Sample

J55 steel specimens were used in all experiments. All specimens used for weight-loss experiments were machined to rectangle coupons. Before the initiation of the experiment, the metal surface was mechanically grounded with emery papers graded 400–1200, washed with acetone and double distilled water, and lastly dried using a dryer. The J55 steel composition was (wt %): C (0.31), Mn (0.92), Si (0.19), P (0.01), Cr (0.2), S (0.008), and Fe in balance. The dimension of the steel coupons used for weight-loss and electrochemical experiments were 5.0 cm × 2.5 cm × 0.2 cm and 1 cm^2^.

### 2.2. Corrosive Medium

The corrosive medium (3.5 wt % NaCl) was prepared using analytical grade NaCl and double distilled water. The concentration ranges of each tested inhibitor used in the course of the experiments were 50 to 400 mg/L. Before the experiments, N_2_ gas was bubbled for 3 h in the corrosive solution in order to remove the oxygen. Then, the solution was deoxygenated by purging CO_2_ gas for 4 h. The specimens were then immersed into the solution while the CO_2_ gas-purging at a pressure of 6 MPa was maintained to ensure a full saturation throughout the test. The electrochemical setup was sealed during the experiment. The initial pH of the corrosive medium was 4.

### 2.3. Synthesis of Inhibitor

The synthesis of benzimidazole derivatives was carried out using the reported method [16]. In a round bottom flask, aromatic aldehyde (2 mmol), o-phenylenediamine (2 mmol), boric acid (0.1 g), and water (10 mL) were stirred at room temperature for 15–30 min. After the completion of the reaction, 5 mL water was added and the mixture was further stirred for 10 min. The obtained precipitate was filtered and purified by recrystallization from ethanol. The characterization of benzimidazole derivatives was done by ^1^H NMR and ^13^C NMR (AVH D 500 ADVANCE III HD One Bay NMR Spectrometer, Bruker Bio Spin International AG, Billerica, MA, USA). ^1^H and ^13^C spectra were recorded at 400 MHz and 100 MHz, respectively, using CDCl_3_ as a solvent. The synthesis scheme and molecular structure are shown in Figure 1. The ^1^H NMR and ^13^C NMR spectra are given in the Appendix A.

### 2.4. NMR Data for Synthesized Inhibitors

#### 2.4.1. 2-(4-Methoxyphenyl)-1*H*-Benzo[*d*]Imidazole (MMI)

^1^H NMR (300 MHz, DMSO-d_6_) δ (ppm): 12.72 (brs, 1H, *NH*), 8.10 (d, *J* = 8.39 Hz, 2H), 7.54 (brs, 2H), 7.19–7.07 (m, 4H), 3.82 (s, 3H).

^13^C NMR, δ (ppm): 55.38, 114.43, 122.34, 123.54, 128.58, 151.58, 161.78.

#### 2.4.2. 2-(3,4-Dimethoxyphenyl)-1*H*-Benzo[*d*]Imidazole (DMI)

^1^H NMR (300 MHz, DMSO-d6) δ (ppm): 12.76 (s, 1H, *NH*), 7.80–7.71 (m, 2H), 7.66–7.44 (m, 2H), 7.20–7.06 (m, 3H), 3.87 (s, 3H), 3.82 (s, 3H).

^13^C NMR δ (ppm): 151.52, 150.31, 148.92, 143.76, 134.95, 122.74, 121.85, 119.30, 118.22, 111.81, 111.08, 109.71, 55.62, 55.59.

#### 2.4.3. 2-(3,4,5-Trimethoxyphenyl)-1*H*-Benzo[*d*]Imidazole (TMI)

^1^H NMR (300 MHz, DMSO-d6) δ (ppm): 12.87 (s, 1H, *NH*), 7.71–7.48 (m, 5H), 7.19 (s, 2H), 3.88 (s, 6H), 3.69 (s, 3H).

^13^C NMR δ (ppm): 153.32, 151.33, 143.79, 138.87, 135.00, 125.59, 122.66, 121.82, 118.78, 111.36, 103.81, 60.31, 56.12.

### 2.5. Weight Loss

ASTM G31-2004 standard was used to determine the duration of the test for the weight-loss experiments. The test duration was 24 h. The corrosion rates were calculated using the weight-loss method data. After 24 h of exposure, the steel coupons were rinsed with distilled water and Clarke’s solution for 5 min. Finally, the coupons were rinsed with distilled water and dried. All weight-loss experiments were performed in triplicate at 333 K for 24 h. Through the weight-loss method, the corrosion rates were calculated as follows:(1)$$C_{R}=\frac{8.76\times \Delta m}{\rho at}$$
where *W* is the average weight loss of the J55 steel specimen (mg), *a* is total area of the J55 steel specimen, *t* is the immersion time (24 h), and *D* is density of the J55 steel in (g cm^−3^).

### 2.6. Electrochemical Analysis

An Autolab Potentiostat device (Metrohm, the Netherland) was used for electrochemical analysis. A three electrode setup was attached to the potentiostat that had a saturated calomel electrode (SCE) as a reference electrode, a graphite rod as an auxiliary electrode, and the J55 steel as the working electrode. At first, the working electrode was immersed in the test medium, i.e., 3.5% NaCl saturated with carbon dioxide for 30 min at 303 K before each experiment to maintain the steady state corrosion potential (*E_corr_*).

The electrochemical impedance spectroscopy (EIS) was performed in the range of frequency 100 kHz to 10 mHz at an open circuit potential, by setting 10 mV as the AC sine wave amplitude frequency per decade. Calculations of inhibition efficiencies were done as follows:(2)$$\eta \%=\left(1-\frac{R_{ct}}{R_{ct(i)}}\right)\times 100$$
where *R_ct_* and *R*_*ct*(*i*)_ are representative of the resistance of charge transfer without and with the studied inhibitors, respectively.

The potentiodynamic polarization study of the J55 steel without and with the inhibitors was conducted in the range of −250 mV to +250 mV potential and the scan rate used was 1 mV/s. The following equation was used for the inhibition efficiency calculation:(3)$$\eta \%=\left(1-\frac{i_{corr(i)}}{i_{corr}}\right)\times 100$$
where, *i_corr_* and *i_corr*(*i*)*_* represent the values of corrosion current densities without and with inhibitors, respectively.

### 2.7. X-ray Photoelectron Spectroscopy (XPS)

XPS (VG ESCALAB 220 XL spectrometer instrument, Thermo Scientific, Waltham, MA, USA) was used to analyze the chemical composition of corrosion products on the specimen after testing in the test solution. The processing of XPS data was achieved using XPS Peak-Fit 4.1 software (Hong Kong, China). The high resolution XPS spectra of C 1s, N 1s, O 1s, and Fe 2p of the TMI inhibitor were analyzed.

### 2.8. Quantum Chemical Calculation

The quantum chemical calculation was performed using density functional theory (DFT). The basis sets used in the present investigation were the DFT/B3LYP methods using 6-311G (d, p) and the Gaussian 09 program package (Wallingford, CT, USA) [17].

### 2.9. MD Simulations and Radial Distribution Function

BIOVIA Materials Studio software 7.0 (San Diego, CA, USA) were used for simulations [18]. A slab size of the 5 Å Fe (110) surface was selected due to its packed and stable configuration [19]. To allow for better metal-inhibitor interaction-analysis, a simulation box with dimensions of 24.82 × 24.82 × 35.69 Å^3^ was used. Also in the simulation box, corrosive particles such as 9Cl^−^, 491H_2_O, 9${\mathrm{CO}}_{3}^{2-}$, and benzimidazole derivatives in their neutral and protonated forms were added. All simulations were executed at a temperature of 303 K and an Andersen thermostat was used to maintain the constant temperature. A COMPASS force field was used for energy minimization and MD calculation processes [20,21].

The radial distribution functions (RDFs) can be defined as the probability of finding particle B around particle A at a definite range. The RDF calculations were performed using simulation trajectories [22]. The radial distribution function can be represented as [23]
(4)$$g_{AB}(r)=\frac{1}{{\langle \rho_{B}\rangle }_{\mathrm{local}}}\times \frac{1}{N_{A}}\sum_{i\in A}^{N_{A}}\sum_{j\in B}^{N_{B}}\frac{\delta (r_{ij}-r)}{4\pi r^{2}}$$
where *ρ_B_*_local_ is the density of particle *B* averaged over all shells around particle *A*.

## 3. Results and Discussion

### 3.1. Weight-Loss Experiment

#### 3.1.1. Concentration Effect

The effect of the benzimidazole derivative concentrations on the protection ability of the metal surface is represented in the form of a concentration vs. inhibition efficiency plot (Figure 2a) Table 1.

From Figure 2a–c and Table 1, it is obvious that the inhibitory performance of benzimidazole derivatives increased with an increase in their concentration and attained values of 94% (TMI), 83.7% (DMI), and 79% (MMI) at 400 mg/L. The increase in inhibitory performance is due to more coverage of the metal surface because of the adsorption of the benzimidazole derivatives molecules onto the J55 steel surface, which finally reduced the attack of acid. This finding suggests that the molecular structure of benzimidazole derivative molecules has a great influence on the inhibition efficiency values. In this study the inhibitor molecules had π electrons in the aromatic ring and non-bonding electrons on the heteroatoms, such as oxygen and nitrogen, which helped the inhibitor to adsorb onto the J55 steel surface [24].

The number of electron donating functional groups could also affect the adsorption tendency of the inhibitor, i.e., with more electron donating functional groups, the adoption would be stronger and the inhibition efficiency would be higher. Thus, in the present case, TMI had the highest protection ability due to the presence of three OCH_3_ groups. Therefore, the inhibition efficiency order was TMI > DMI > MMI.

#### 3.1.2. Adsorption Isotherm of Inhibitor on J55 Steel

Many adsorption isotherms like Langmuir, Frumkin, Flory Huggins, and Temkin were investigated to find a good fit with the experimental study. Out of these isotherms, the Langmuir isotherm, i.e., *C_inh_*/*θ* vs. the inhibitor concentration (*C_inh_*), was found to be the best fit due to the slope and regression coefficient (R^2^) values approaching towards unity (Figure 2b). The Langmuir isotherm is given by the following formula [25]:(5)$$\frac{C_{inh}}{\theta }=\frac{1}{K_{ads}}+C_{inh}$$
where *C_inh_* is the benzimidazole derivatives concentration (mg/L) and *θ* and *K_ads_* represent the surface coverage and equilibrium adsorption constant, respectively. Although examination of slope values suggests a good fit, it slightly deviates from unity, which is not consistent with the Langmuir adsorption isotherm assumption of monolayer adsorption of inhibitor molecules on the metal surface. According to Eduok and Khaled [26], the discrepancies in slope values are related to the adsorption phenomena, and thus it is important to consider another physical characteristic of the adsorption isotherm. The Langmuir adsorption isotherm can be mathematically represented in terms of the dimensionless separation constant (*K_L_*), and is given by the following equation [26]:(6)$$K_{L}=\frac{1}{1+K_{ads}C}$$
where *K_L_* is the dimensionless separation factor of inhibitor-adsorption. The mean values of the calculated *K_L_* are given in Table 2. Ideally, when the value of *K_L_* is less than unity, the adsorption process is considered to be favorable and the experimental data fit the Langmuir adsorption isotherm. The adsorption process is unfavorable when *K_L_* is greater than unity, and irreversible at *K_L_* = 1. The mean values of *K_L_* were less than unity, suggesting that the adsorption process was favorable. The Appendix A contains the Frumkin, Flory Huggins, and Temkin isotherm plots. The *K_ads_* values were determined from the intercept of the Langmuir plots and are listed in Table 2. The strength of the adsorption of the benzimidazole derivatives molecules on J55 steel are represented by the values of *K_ads_*. From the table it can be observed that as the value of the OCH_3_ group increased, the values of *K_ads_* increased, and the highest *K_ads_* was for TMI, which suggests it is had the strongest adsorption onto the metal surface [27,28].

#### 3.1.3. Thermodynamic Parameters of Adsorption

The adsorption behavior and the nature of the adsorption were determined by calculating the thermodynamic parameters of adsorption. The standard free energy of adsorption, i.e., Δ$G_{ads}^{o}$, was correlated to *K_ads_* according to the following equation [29]:(7)$$\Delta G_{ads}^{o}=-2.303RT\log(55.5K_{ads})$$
where the absolute temperature and universal gas constants are represented by *T* and *R,* respectively, and 55.5 is the magnitude of the water molecules concentration. Table 2 reveals that the Δ$G_{ads}^{o}$ values are negative, suggesting a spontaneous adsorption process [30].

Thermodynamically, *K**_ads_* is related to the standard enthalpy and entropy of adsorption, i.e., ∆$H_{ads}^{o}$ and ∆$S_{ads}^{o}$ and can be calculated using the Van’t Hoff equation:(8)$$\ln K_{ads}=\frac{-\Delta H_{ads}^{o}}{RT}+\frac{\Delta S_{ads}^{o}}{R}-\ln C_{H_{2}O}$$
where ∆$H_{ads}^{o}$ and ∆$S_{ads}^{o}$ are the standard enthalpy and entropy of adsorption. The graph of ln *K*_ads_ vs. 1/*T* is given in Figure 2c. The slopes of (∆$H_{ads}^{o}$/*R*) were used for the calculation of ∆$H_{ads}^{o}$ values and are shown in Table 2. In general, the adsorption process is exothermic in nature, which means it is accompanied by the release of energy. All values of ∆$H_{ads}^{o}$ were negative, so the process of adsorption of inhibitor molecules was exothermic in nature. It should be noted that in the present investigation, ∆$H_{ads}^{o}$ values were between −52 and −81 kJ/mol, which reveals that the adsorption of the benzimidazole derivatives was both physical and chemical in nature [31]. One more important parameter is that Δ$G_{ads}^{o}$ was between −40 kJ/mol and −20 kJ/ mol, which further confirms that the adsorption of inhibitor molecules on the J55 steel surface was both physical and chemical [32,33,34,35].

### 3.2. Electrochemical Studies

#### 3.2.1. Electrochemical Impedance Spectroscopy (EIS) Studies

The capacitive and inductive behaviors of J55 steel were studied by an electrochemical impedance study. The impedance behavior of the metal is represented in the form of Nyquist plots (Figure 3a–c) at 308 K. According to Figure 3a–c, at a high frequency, a capacitive loop arises that consists of a depressed semicircle. This phenomenon is due to the charge-transfer and capacitance created by the double layer [36,37]. Also, the capacitive loop diameter in the presence of benzimidazole derivatives is larger as compared to that of the blank. Meanwhile, the capacitive loop diameter increased as the benzimidazole concentration increased, which is because of the adsorption of the benzimidazole derivative molecules onto the metal, which formed a barrier of inhibitor molecules and, in turn, enhanced the corrosion-resistance property of the metal [38,39,40].

It is interesting to note that in the absence of inhibitors and at 100 mg/L concentration of inhibitors, an inductive loop was also observed due to the adsorption of the intermediate product, i.e., FeOH*_ads_*, which was formed during the dissolution of the J55 steel [41]. However, as the concentration of benzimidazole derivatives increased, only capacitive loops were observed and inductive loops disappeared. This may be due to a larger area of the J55 steel being covered by benzimidazole derivative molecules with increasing concentration, which finally reduces the J55 steel corrosion.

For impedance data calculations, two circuits were used and are shown in Figure 3d,e. The elements used for construction of the circuits consisted of a resistor for charge transfer (*R_ct_*), inductor (*L*), solution resistor (*R_s_*), and constant phase element (CPE). The CPE was used for accurate fitting of the circuit because it can compensate for the effects that cause deviations such as surface roughness, dislocation, imperfection, impurities, inhibitor adsorption, etc. [42,43,44,45,46].

Table 3 represents the fitted-curve impedance results. It can be noted that the *R_ct_* and *Y*_0_ values at each concentration of all benzimidazole derivatives are showing an opposite trend, in other words, *Y*_0_ decreases and *R_ct_* increases. This phenomenon is because of the adsorption of the benzimidazole molecules onto the J55 steel, which finally enhances the J55 steel corrosion-resistance property [47]. The values of *R_ct_* at 400 mg/L for TMI, DMI, and MMI were 2466 Ω cm^2^, 866 Ω cm^2^, and 697 Ω cm^2^, respectively. Thus, the TMI derivative provided the maximum resistance towards corrosion. This is because of the three electron donating OCH_3_ groups in TMI. The increase in *n* values with the addition of bezimidazole derivatives is due to the adsorption of the derivatives, which enhances the homogeneity of the derivatives [48]. Thus, with an increase in the number of electron donating functional groups, the corrosion inhibition property is increased. Therefore, the effectiveness of the benzimidazole derivatives as corrosion inhibitors can be given as TMI > DMI > MMI.

#### 3.2.2. Potentiodynamic Polarization Analysis

The Tafel plots of the J55 steel in 3.5% NaCl saturated with carbon dioxide without and with different concentrations of benzimidazole derivatives are represented in Figure 4a–c. Some important electrochemical parameters such as corrosion current density (*i_corr_*), corrosion potential (*E_corr_*), cathodic Tafel slope (*β*_c_), anodic Tafel slope (*β*_c_), and inhibition efficiency (IE%) are shown in Table 4. The observation of Table 3 shows that the corrosion-current density values shift from 104.4 μA/cm^2^ (Blank) to 5.5 μA/cm^2^ (TMI), 13.2 μA/cm^2^ (DMI), and 21.5 μA/cm^2^ (MMI), and this represents that benzimidazole derivatives are effective corrosion inhibitors. The values of the maximum inhibition efficiency are 94%, 87%, and 79% for TMI, DMI, and MMI, respectively at 400 mg/L. Also, the addition of the benzimidazole derivative molecules to the aggressive medium caused reduction of both the anodic and cathodic current density. All the studied benzimidazole derivative molecules with the increment of concentration reduced more H^+^ ions in the cathodic reactions as compared to anodic dissolution reactions. According to Figure 4a–c, the cathodic Tafel lines are parallel, suggesting that the activation-control evolution of H_2_ gas and the mechanism of H^+^ to H_2_ conversion/reduction are not modified by the presence of benzimidazole derivatives. Also, with increased concentration of benzimidazole derivatives, the *β*_c_ values were changed, indicating that benzimidazole derivatives affected the hydrogen evolution kinetics. This is because of the diffusion or barrier effect [49]. Similarly, the slope values of the anodic Tafel lines, i.e., *β*_a_, also underwent changes with increased benzimidazole derivative concentration, suggesting that the studied inhibitor molecules were initially adsorbed over the J55 steel surface, reducing the process of corrosion by blocking the reactive sites presented on the J55 steel without altering the mechanism of the anodic reaction [50].

Tafel curves showed that the addition of benzimidazole derivatives inhibited both the cathodic and anodic reactions. Thus, the inhibitor is said to be a mixed type. However, the shift in the *E_corr_* values in the inhibited solution was towards the cathodic direction, i.e., negative with respect to the uninhibited solution, revealing that while benzimidazole derivatives were predominantly cathodic, overall they were a mixed-type inhibitor. It is interesting to note here that as the number of OCH_3_ groups increased, a larger reduction in the corrosion current density occurred. This is due to the donation of an electron by the OCH_3_ group, which facilitates formation of stronger bonds between benzimidazole derivative molecules and the J55 steel. Thus, the inhibition efficiency order was TMI > DMI > MMI.

### 3.3. X-ray Photoelectron Spectroscopy (XPS)

For the confirmation of TMI derivative adsorption on J55 steel, XPS analyses were performed and elaborated. The XPS spectra obtained for TMI adsorbed on the J55 steel at a concentration of 400 mg/L after 24 h immersion in a corrosive solution are shown in Figure 5a–d. Spectra of the XPS consist of the following peaks; C 1s, N 1s, O 1s, and Fe 2p. All spectra are complex in nature, and in order to assign the peaks of the corresponding adsorbed species, a deconvolution fitting procedure was used.

Three main peaks are in C 1s spectra. Appearance of first peak occurs at the binding energy of 284.79 eV and is attributed to the aromatic bonds between C–C, C=C, and C–H [51,52,53]. The appearance of the second peak occurs at 286.21 eV and corresponds to the carbon atoms bonded to nitrogen in C–N and C=N bonds present in the imidazole ring [51,52,53]. The third peak appears at 288.87 eV and represents the nitrogen bonded carbon atom of the imidazole ring, i.e., C=N+ [54], that results because of protonation of the =N– atom in the imidazole ring and/or the imidazole ring nitrogen coordination with the J55 steel. The N 1s XPS spectrum consists of one peak at 399.27 eV. The appearance of this peak is due to the C–N and the unprotonated N atom (=N– structure) in the imidazole ring [55,56,57].

The spectrum of O 1s has three main peaks. The peak that appears at approximately 530.35 eV is attributed to O^2−^, and it corresponds to the oxides of iron and oxygen, i.e., Fe_2_O_3_ and/or Fe_3_O_4_ [58]. The appearance of the second peak at a binding energy of 531.33 eV is because of OH^−^ and is attributed to the existence of the hydrous form of iron oxides, such as FeOOH [58]. Lastly, oxygen of adsorbed water molecules appears as a third peak at 532.05 eV [57].

The Fe 2p spectrum for the J55 steel surface covered with TMI derivatives consists of two doublets, one at 711.78 eV (Fe 2p_3/2_) and the second at 724.98 eV (Fe 2p_1/2_). The deconvolution of the high resolution Fe 2p3/2 XPS spectrum consists of two peaks. The first peak at the binding energy of 709.4 eV is attributed to metallic iron, i.e., iron in the zero oxidation state [59,60]. The appearance of the second peak at 711.78 eV is due to Fe^3+^ [61] and is attributable to the oxides of iron such as Fe_2_O_3_, Fe_2_CO_3_, and FeOOH (i.e., oxyhydroxyde) [62,63]. The comparison between Fe 2p_3/2_ XPS results of the TMI treated steel with that of untreated steel (as described previously) [64] shows that there is a significant decrease in the amount of Fe^0^ that indicates an increment in the oxide layer thickness. The formed oxide layer of FeOOH is insoluble and stable, which reduces the diffusion of metal ions and thus enhances the corrosion-resistive property of J55 steel in an aggressive media.

### 3.4. Quantum Chemical Calculation

#### Calculation of Preferred Site for Protonation

In an aqueous medium, organic molecules can easily undergo protonation, and this protonated form of the inhibitor takes part in the adsorption process. In the present case, the number of nitrogen atoms in the imidazole ring with the most negative Mullikien charge is two (Table 5). Therefore, there are protonated N_2_ and N_5_ nitrogen atoms. However, the most preferential nitrogen atom that can undergo protonation was selected by calculating the proton affinity (*PA*) at both N_2_ and N_5_. The equation used for calculating *PA* is given below:(10)$$PA=E_{prot}-\left(E_{neutral}+E_{H^{+}}\right)$$
(11)$$E_{H^{+}}=E_{H_{3}O^{+}}-E_{H_{2}O}$$
where the total energies of the protonated inhibitor forms and the neutral inhibitor forms are represented by *E_prot_* and *E_neutral_*, respectively. *E_H_*2*_O_* is the water molecule total energy and *E*_*H*_3_*O*^+^_ is the hydronium ion total energy. The site having the most negative value of *PA* is selected as the most preferable site for protonation. Thus, in this paper, the calculated value of PA for N_5_ was the most negative and so it was selected as the preferential site for protonation (Table 5).

### 3.5. Molecular Dynamic Simulations

Despite extensive research having been conducted in recent years, there is uncertainty regarding the corrosion inhibition mechanisms of CO_2_ corrosion, and more critical investigations should be conducted. In this regard, MD simulations could be a good way to improve scientific knowledge in the field. In the course of this study, we investigated the adsorption of neutral and protonated inhibitor molecules on an iron surface in the presence of a simulated electrolyte with the aim of mimicking the experimental conditions and assessing whether any relationship exists between the theoretical and experimental results, and, if so, how significant the MD results are in explaining the inhibition process. Simulations were run until the systems reached an equilibrium state, then, the interaction energies were estimated by calculating the single point energies of all system constituents [65]. The obtained equilibrium configurations of neutral and protonated forms of inhibitor molecules on the Fe (110) surface in solution are shown in Figure 6 and Figure 7. It can be observed from the results in Figure 6 and Figure 7 that both forms of the inhibitor molecules are adsorbed on the iron surface in a parallel manner and are near to the iron surface. Such situations can help to produce chemical interactions and thereby increase the adsorption rate of tested inhibitors. The benzimidazole itself is a good corrosion inhibitor, thus, with the addition of a methoxy group in the phenyl ring, the interactive forces and the affinity toward the J55 steel of our compounds strongly increased, which lead to more interactions with the steel surface [66].

The interaction and binding energies (*E*_Binding_ = −*E*_interaction_) of the obtained inhibitor molecules under equilibrium conditions for neutral and protonated forms can also be useful information to assess the extent of adsorption of the three compounds. The results in Table 6 show higher energy values that may explain the higher interaction between the inhibitor molecules and the steel surface and the stability of formed films [67,68]. The energy values are slightly decreased in a protonated form that can be explained mainly by the higher contribution of the physical interactions between protonated inhibitor molecules and the positively charged metal surface. We can also note that the energy values follow the same trend of the inhibition efficiency values, thus confirming the crucial role of the number of methoxy groups as powerful electro donating groups in increasing the adsorption abilities of the tested compounds.

### 3.6. Radial Distribution Function (RDF)

RDF analysis provides further insights into the interactive force of an inhibitor molecule and its affinity towards the iron surface [23]. Here, the total radial distribution function was calculated for both inhibitor forms using MD simulation trajectories. Whether the interactions of an inhibitor with iron atoms are meaningful can be judged by comparison of the first prominent peaks in the RDF curves. If the peak occurs at 1 Å ~ 3.5 Å, it is an indication of a small bond length, which correlates to chemisorption, while the physical interactions are associated with the peaks longer than 3.5 Å [69]. Figure 8 shows the RDF results of neutral and protonated forms. We can see that the first prominent peak for both inhibitor forms is located at a distance smaller than 3.5 Å. From Figure 8, one can easily observe that the first peak increased with the decreased inhibition efficiency of the tested compounds. A further increase was observed in the protonated state of the inhibitor molecules. All the inhibitor molecules in their neutral or protonated forms retained significant interaction with the iron surface.

## 4. Conclusions

(1)The tested benzimidazole derivatives are good inhibitors in the aggressive media of 3.5% NaCl solution saturated with carbon dioxide at 333 K.(2)Experimental and theoretical investigations suggest that as the number of methoxy groups increase so too does the corrosion protection ability of the inhibitors, and thus TMI is the best inhibitor.(3)The potentiodynamic polarization measurement supports the mixed mode of inhibitors with predominantly cathodic effects.(4)Langmuir adsorption is the preferred isotherm for all inhibitors.(5)XPS micrographs support the benzimidazole derivative adsorption.(6)The DFT study confirms that the imine nitrogen (N_5_) is the most preferred site for protonation.(7)MD results support that TMI has a stronger adsorption ability than that of both DMI and MMI.(8)Results of the RDF study confirmed that both the neutral and protonated form of the inhibitor show significant interaction with the steel surface.

## Figures and Tables

**Figure 1 materials-12-00017-f001:**
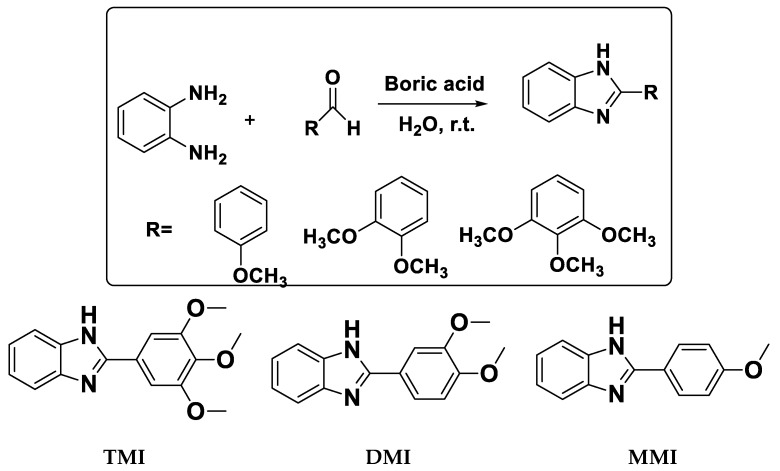
Synthetic scheme and molecular structure of the benzimidazole derivatives. 2-(3,4,5-Trimethoxyphenyl)-1*H*-benzo[*d*] imidazole (TMI); 2-(3,4-Dimethoxyphenyl)-1*H*-benzo[*d*] imidazole (DMI); and 2-(4-Methoxyphenyl)-1*H*-benzo[*d*] imidazole (MMI).

**Figure 2 materials-12-00017-f002:**
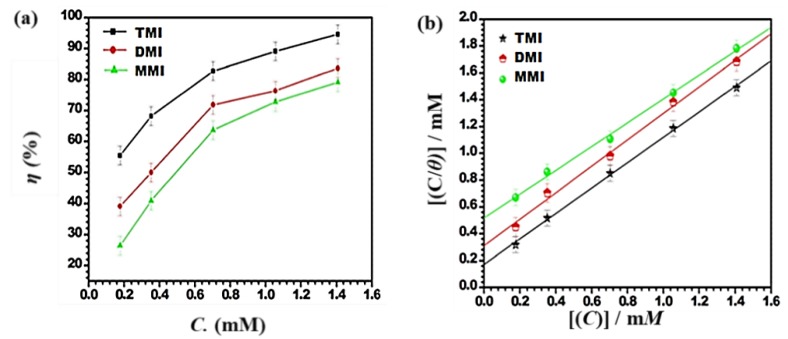
(**a**) Variation of inhibition efficiency (*η* %) with inhibitor concentration at 333 K; (**b**) Langmuir Isotherm plots for adsorption of inhibitors; (**c**) The relationship between ln *K_ads_* and 1000/*T* at optimum concentration of inhibitors.

**Figure 3 materials-12-00017-f003:**
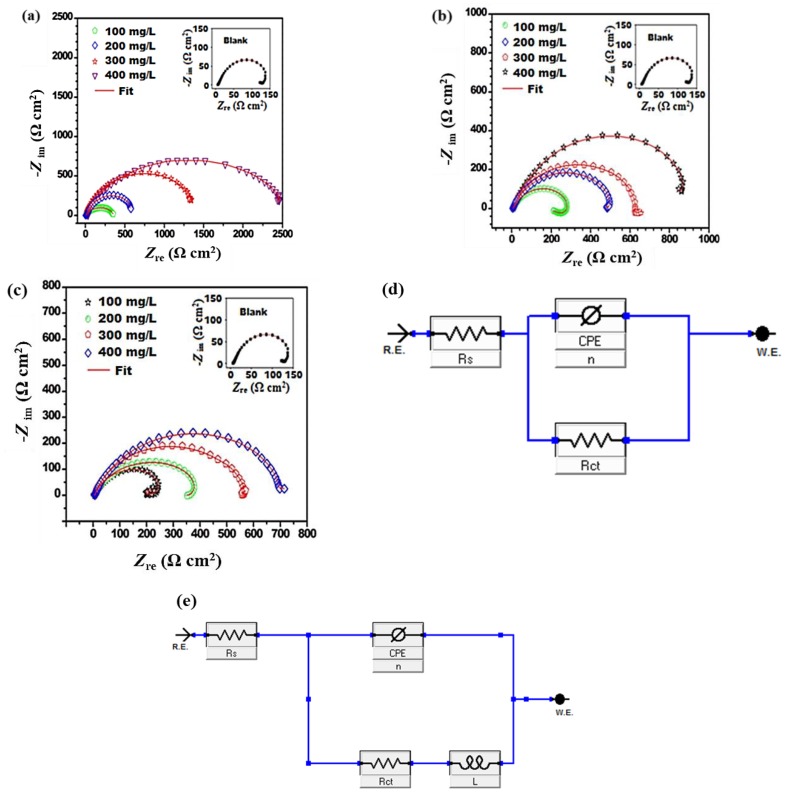
Nyquist plots in absence and presence of different concentration of inhibitors: (**a**) TMI; (**b**) DMI; (**c**) MMI; (**d**,**e**) Equivalent circuits used.

**Figure 4 materials-12-00017-f004:**
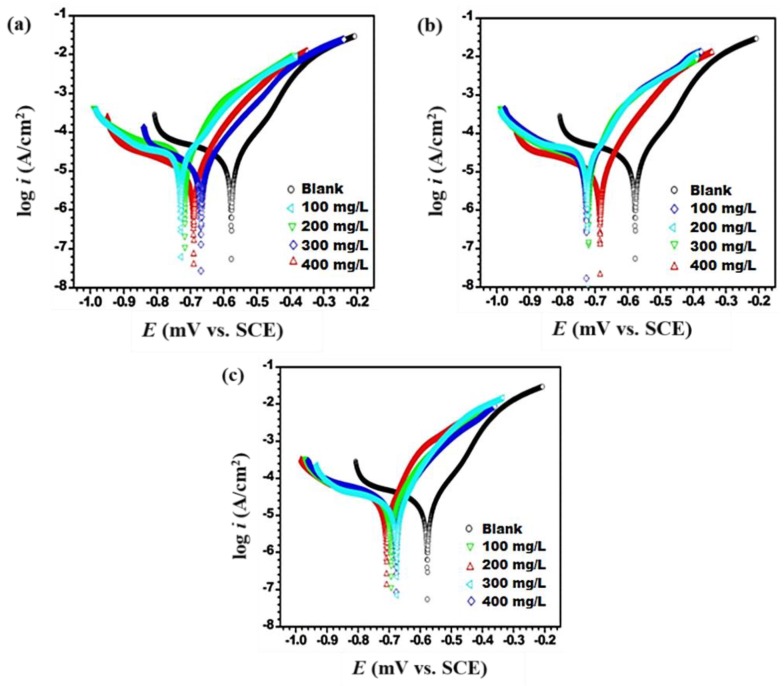
Potentidynamic polarization curves in absence and presence of different concentrations of inhibitors: (**a**) TMI; (**b**) DMI; (**c**) MMI.

**Figure 5 materials-12-00017-f005:**
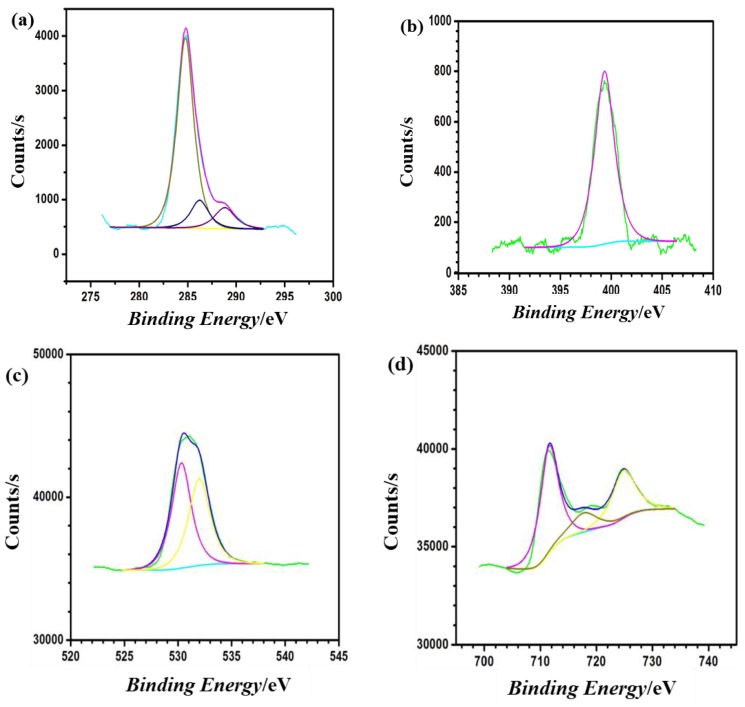
XPS spectra: (**a**) C 1s, (**b**) N 1s, (**c**) O 1s and (**d**) Fe 2p of the TMI inhibitor.

**Figure 6 materials-12-00017-f006:**
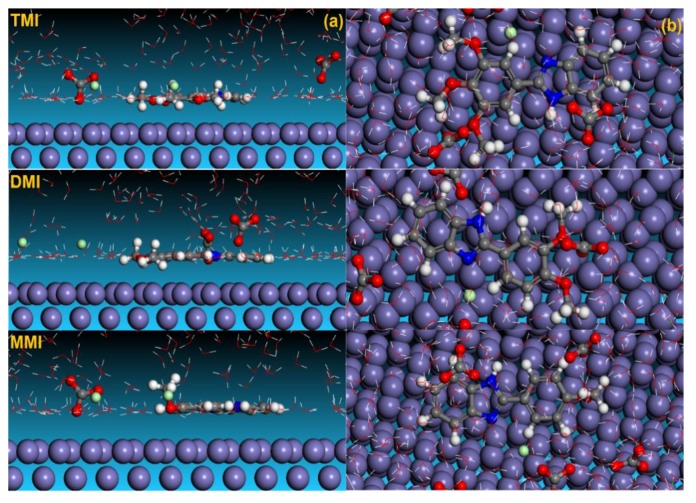
Side and top views of the final adsorption of neutral forms of inhibitor molecules on the Fe (110) surface in solution: (**a**) side view; (**b**) top view.

**Figure 7 materials-12-00017-f007:**
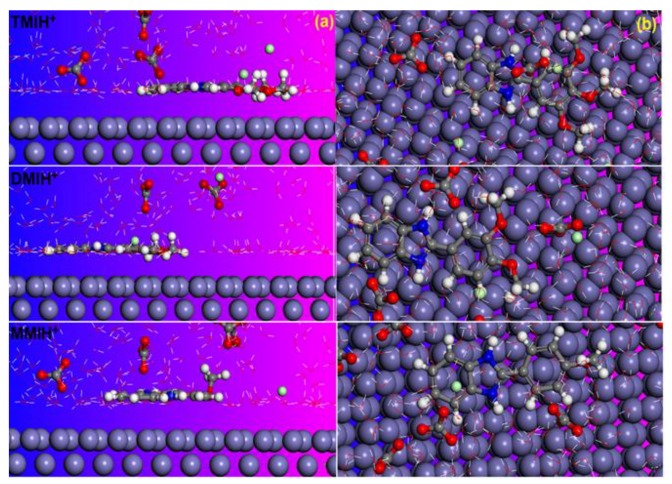
Side and top views of the final adsorption of the protonated forms of the inhibitor molecules on the Fe (110) surface in solution: (**a**) side view; (**b**) top view.

**Figure 8 materials-12-00017-f008:**
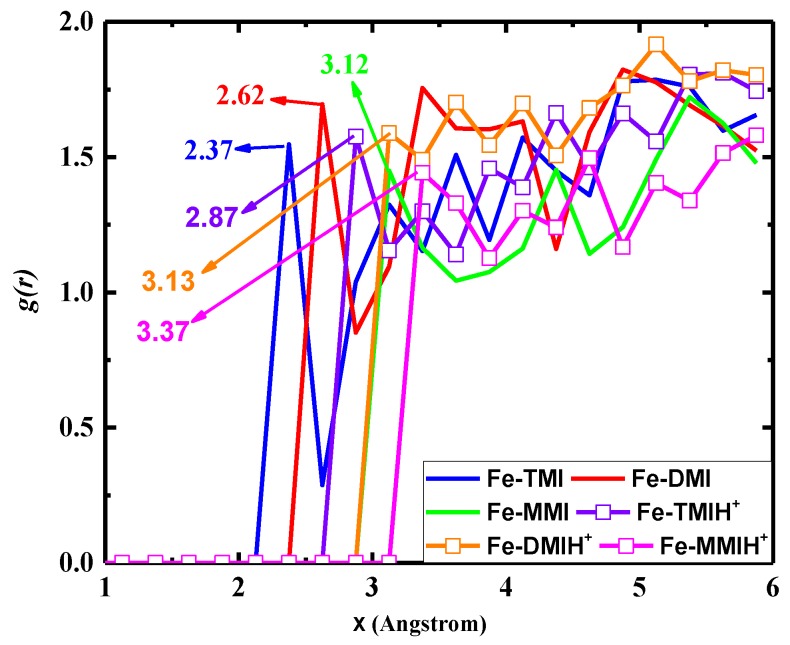
Radial Distribution Functions (RDFs) of neutral and protonated forms of the tested corrosion inhibitors adsorbed on the Fe (110) surface in solution.

**Table 1 materials-12-00017-t001:** Corrosion inhibition efficiency with the inhibitor concentrations.

Concentrations (mM)	*η* (%)		
	TMI	DMI	TMI
0.176	55.4	39.0	26.3
0.352	68.1	50.0	40.9
0.703	82.7	71.8	63.6
1.055	89.0	76.3	72.7
1.407	94.5	83.6	79.0

**Table 2 materials-12-00017-t002:** Langmuir adsorption isotherm and thermodynamic parameters for the synthesized inhibitors.

Inhibitor	Slope	Regression Coefficient (R^2^)	*K_ads_* (10^3^ M^−1^)	$\Delta G_{ads}^{o}$ (kJ/mol)	$\Delta H_{ads}^{o}$ (kJ/mol)	*K_L_* (mean)
TMI	0.949	0.998	5.88	−35.15	−81.87	0.251
DMI	0.988	0.995	3.22	−33.49	−57.59	0.364
MMI	0.888	0.997	1.93	−32.07	−52.74	0.472

**Table 3 materials-12-00017-t003:** Electrochemical impedance parameters in absence and presence of different concentrations of inhibitors at 308 K.

*C_inh_* (mg L^−1^)	*R_s_* (Ω)	*R_ct_* (Ω cm^2^)	*Y*_0_ (μF/cm^2^)	*n*	*L* (*H*)	*η* (%)
Blank	4.817	135.57	512.7	0.601	8.04	--
**TMI**						
100	4.927	350.33	278.3	0.789	--	61.3
200	5.190	573.21	210.6	0.854	--	76.3
300	5.562	1400.01	140.9	0.855	--	90.3
400	5.601	2466.17	42.45	0.879	--	94.5
**DMI**						
100	6.159	269.84	296.1	0.771	60.02	49.7
200	5.930	485.34	285.9	0.816	--	72.0
300	5.593	624.24	156.5	0.839	--	78.2
400	5.473	866.87	76.56	0.848	--	84.3
**MMI**						
100	5.255	231.39	325.2	0.769	35.46	41.4
200	6.01	377.24	291.4	0.804	--	64.0
300	5.416	557.11	234.5	0.812	--	75.6
400	5.601	697.71	125.5	0.827	--	80.5

**Table 4 materials-12-00017-t004:** Electrochemical polarization parameters in the absence and presence of different concentrations of inhibitors at 308 K.

Inhibitor	*E_corr_* (mV/SCE)	*i_corr_* (μA/cm^2^)	*β*_a_ (mV/dec)	−*β*_c_ (mV/dec)	*η* (%)
Blank	−576	104.4	154	590	--
**TMI**					
100	−666	45.0	192	683	56.8
200	−717	29.3	92	443	71.9
300	−669	20.0	114	499	80.7
400	−691	5.5	80	382	94.7
**DMI**					
100	−727	51.4	190	693	50.5
200	−723	34.4	106	425	66.9
300	−720	25.2	82	489	75.7
400	−685	13.2	85	499	87.2
**MMI**					
100	−694	65.2	183	796	37.2
200	−709	45.9	118	846	55.8
300	−678	27.7	105	542	73.2
400	−676	21.5	183	511	79.3

**Table 5 materials-12-00017-t005:** Atomic charges on hetroatoms and proton affinity values. *PA* = proton affinity.

Inhibitors	N_2_	N_5_	*PA* (kcal/mol)	
			N_2_	N_5_
TMI	−0.495	−0.402	5.65	−30.75
DMI	−0.492	−0.403	6.27	−30.12
MMI	−0.490	−0.403	5.65	−29.34

**Table 6 materials-12-00017-t006:** Selected energy parameters obtained from molecular dynamic (MD) simulations for adsorption of inhibitors on the Fe (110) surface.

System	Neutral Form	Protonated Form
	*E*_interaction_ (kJ/mol)	*E*_interaction_ (kJ/mol)
Fe + TMI	−564.09	−559.77
Fe + DMI	−507.34	−497.19
Fe + MMI	−453.67	−444.31

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
