# Peer review of "Effect of Electron Donating Functional Groups on Corrosion Inhibition of J55 Steel in a Sweet Corrosive Environment: Experimental, Density Functional Theory, and Molecular Dynamic Simulation"

_materials, 2018, doi:10.3390/ma12010017_

Round 1
Reviewer 1 Report
I have reviewed the article entitled " Effect of electron donating functional groups on corrosion inhibition of J55 steel in sweet corrosive environment: Experimental, density functional theory and molecular dynamic simulation.
The paper deals with the impact of synthesized benzimidazole derivatives on the corrosion of J55 steel immersed in CO2-saturated chloride solution. For this purpose, electrochemical characterizations are conducted. The study is clear and appropriately conducted. The results are interesting and the discussion is supported by appropriate references.
Therefore, I recommend this article for publication after considering the minor revision(see PDF file)

Author Response
Response to reviewer
PDF comments
Query: I am confused. Here is 333 K and in line 215 is 308 K.
Answer: Weight loss experiment was done at 333 K and electrochemical at 308 K.
Query:Did you increase the temperature to 333K after this 30 minutes?
Answer:Working electrode was immersed in the corrosive medium for 30 min at 303 K before each experiment in order to maintain the steady state corrosion potential (Ecorr).
Query: The values of Kads as well as all others parameters (e.g. DGads, inhibition efficiency etc) calculated at different temperatures must also be reported in table 1
the authors may not leave the readers or reviewers to judgement of validity of data for the graphic alone
Answer: A additional table has been added in weight loss section.
Query:Here is 308 K but above was 333 K
Answer:Electrochemical experiment was performed at 308 K
Query:You did not finish the sentence.
Answer: The sentence has been completed
Query:ω =2πfmax fmax represents the frequency at which imaginary value reaches a maximum on the Nyquist plot. I f you want to use this formula you must also report the fmax on the Nyquist plots in figure 3
Answer: The equation has been deleted
Query: In the test the temperature is reported in Kelvin
Answer: The unit has been corrected.
;
Reviewer 2 Report
Manuscript ID: Materials-410367
Title: Effect of electron donating functional groups on 3 corrosion inhibition of J55 steel in sweet corrosive 4 environment: Experimental, density functional theory 5 and molecular dynamic simulation
1. Grammar/spelling: “……stronger adoption ability……” (See concluding section).
2. Grammar: “Then finally the coupons were rinsed with distilled water and then……..” (See Experimental section).
3. I strongly recommend that authors should add relevant related literature of this subject within the introductory section. Below are some relevant published works on the anti-corrosion potentials of Benzimidazoles:
· Journal of Molecular Liquids 246 (2017) 66–90.
· Journal of Electroanalytical Chemistry 655 (2011) 164–172.
4. Section 2.2.: The corrosive medium is a CO2 enriched saline (NaCl) electrolyte. What steps did authors take to ensure that CO2 remained saturated within the medium throughout the duration of the test?
5. Langmuir adsorption isotherm: Langmuir adsorption isotherm assumes equivalence of adsorption sites for inhibitor molecules on the metal surface during binding and demonstrates that inhibitor binding occurs independently of whether nearby sites at the metal–solution interfaces are occupied. Even with good fits, if the slope/gradients of the plots (presented in Table 1) are NOT unity (EQUAL TO 1), this means that the plots are NOT consistent with the Langmuir adsorption isotherm assumption (i.e. monolayer adsorption of inhibitor molecules on the metal surface).
Please, refer to the article below (and many others within the literature) for the reason for this discrepancy and introduction of a correction factor:
· Research on Chemical Intermediates (2015) 41:6309–6324.
·
Author Response
Response to reviewer
Query 1: Grammar/spelling: “……stronger adoption ability……” (See concluding section).
Reply: The spelling error has been corrected.
Query 2: Grammar: “Then finally the coupons were rinsed with distilled water and then……..” (See Experimental section).
Reply 2: The grammar error has been rectified.
Query 3: I strongly recommend that authors should add relevant related literature of this subject within the introductory section.
Reply 3: The relevant references as per your suggestions have been added.
Query 4: Section 2.2.: The corrosive medium is a CO2 enriched saline (NaCl) electrolyte. What steps did authors take to ensure that CO2 remained saturated within the medium throughout the duration of the test?
Reply 4: The CO2 gas was purged till the pH was 4.6~4.8 to ensure a full saturation throughout the test. Then the setup was sealed with epoxy resin to keep the solution saturated for all the tests during the experiment.
Query 5: Langmuir adsorption isotherm: Langmuir adsorption isotherm assumes equivalence of adsorption sites for inhibitor molecules on the metal surface during binding and demonstrates that inhibitor binding occurs independently of whether nearby sites at the metal–solution interfaces are occupied. Even with good fits, if the slope/gradients of the plots (presented in Table 1) are NOT unity (EQUAL TO 1), this means that the plots are NOT consistent with the Langmuir adsorption isotherm assumption (i.e. monolayer adsorption of inhibitor molecules on the metal surface).
Reply 5: The discrepancies in Langmuir plots were corrected and explained in the manuscript.